# Chronic Kidney Disease Transdifferentiates Veins into a Specialized Immune–Endocrine Organ with Increased MYCN-AP1 Signaling

**DOI:** 10.3390/cells12111482

**Published:** 2023-05-26

**Authors:** Fatma Saaoud, Laisel Martinez, Yifan Lu, Keman Xu, Ying Shao, Jia L Zhuo, Avrum Gillespie, Hong Wang, Marwan Tabbara, Alghidak Salama, Xiaofeng Yang, Roberto I. Vazquez-Padron

**Affiliations:** 1Center for Cardiovascular Research, Department of Cardiovascular Sciences, Lewis Katz School of Medicine at Temple University, Philadelphia, PA 19140, USA; 2DeWitt Daughtry Family Department of Surgery, Leonard M. Miller School of Medicine, University of Miami, Miami, FL 33136, USA; 3Tulane Hypertension and Renal Center of Excellence, Department of Physiology, Tulane University School of Medicine, New Orleans, LA 70112, USA; 4Section of Nephrology, Hypertension and Kidney Transplantation, Department of Medicine, Lewis Katz School of Medicine at Temple University, Philadelphia, PA 19140, USA; 5Center for Metabolic Disease Research, Department of Cardiovascular Sciences, Lewis Katz School of Medicine at Temple University, Philadelphia, PA 19140, USA

**Keywords:** chronic kidney disease (CKD), cytokines, chemokines, secretomes, immunometabolism, fibrosis

## Abstract

Most patients with end-stage renal disease (ESRD) and advanced chronic kidney disease (CKD) choose hemodialysis as their treatment of choice. Thus, upper-extremity veins provide a functioning arteriovenous access to reduce dependence on central venous catheters. However, it is unknown whether CKD reprograms the transcriptome of veins and primes them for arteriovenous fistula (AVF) failure. To examine this, we performed transcriptomic analyses of bulk RNA sequencing data of veins isolated from 48 CKD patients and 20 non-CKD controls and made the following findings: (1) CKD converts veins into immune organs by upregulating 13 cytokine and chemokine genes, and over 50 canonical and noncanonical secretome genes; (2) CKD increases innate immune responses by upregulating 12 innate immune response genes and 18 cell membrane protein genes for increased intercellular communication, such as CX3CR1 chemokine signaling; (3) CKD upregulates five endoplasmic reticulum protein-coding genes and three mitochondrial genes, impairing mitochondrial bioenergetics and inducing immunometabolic reprogramming; (4) CKD reprograms fibrogenic processes in veins by upregulating 20 fibroblast genes and 6 fibrogenic factors, priming the vein for AVF failure; (5) CKD reprograms numerous cell death and survival programs; (6) CKD reprograms protein kinase signal transduction pathways and upregulates *SRPK3* and *CHKB*; and (7) CKD reprograms vein transcriptomes and upregulates *MYCN*, *AP1*, and 11 other transcription factors for embryonic organ development, positive regulation of developmental growth, and muscle structure development in veins. These results provide novel insights on the roles of veins as immune endocrine organs and the effect of CKD in upregulating secretomes and driving immune and vascular cell differentiation.

## 1. Introduction

Chronic kidney disease (CKD) affects 11–13% of the global population and 15% of the US population [1,2]. CKD is a debilitating pathology with various causal factors, culminating in end-stage renal disease (ESRD) requiring dialysis or kidney transplantation [3]. CKD is characterized by progressive and irreversible nephron loss, reduced renal regenerative capacity, microvascular damage, metabolic changes, oxidative stress, and inflammation, ultimately resulting in fibrosis [4,5]. Although the majority of CKD patients do not advance to ESRD, current guidelines recommend adequate planning for patients at risk for ESRD to avoid initiating dialysis with a central venous catheter (CVC) [6]. CVC use has been associated with increased mortality, increased hospitalizations, sepsis, and higher healthcare costs compared to other permanent access [7,8]. Hemodialysis vascular access dysfunction is a common and challenging problem in patients with ESRD maintained on chronic hemodialysis, and, indeed, such dysfunction is the single most influential determinant of the rates of hospitalization, morbidity, and mortality in this patient population [9]. Our team previously reported the following: (1) endogenous metabolites, including lysophospholipids [10], can bind to their intrinsic receptors, rather than classical damage-associated molecular pattern (DAMP) receptors, such as toll-like receptors (TLRs) and nod-like receptors/inflammasomes, and become conditional DAMPs; (2) CKD uremic toxins (UTs) serve as conditional DAMPs/homeostasis-associated molecular patterns (HAMPs) to modulate inflammation [11]; (3) upregulated secretomes in peripheral blood mononuclear cells (PBMCs) from patients with CKD and ESRD act synergistically with UTs to promote inflammation and potential disease progression [12]; (4) under pathological conditions, the aorta may act as an immune organ via the upregulation of a variety of secretomes and promote trained immunity [13]; and (5) endothelial cells in the CKD environment can serve as innate immune cells with innate immune memory function by reprograming metabolic pathways [14]. We also reported that CKD contributes to morphometric vascular changes in upper-extremity veins [15]. However, a few important questions remain poorly characterized, including whether CKD pathology contributes to the chronic remodeling and reprogramming of upper-extremity veins into immune endocrine organs and whether CKD primes (trains) upper-extremity veins to fail during the arteriovenous fistula (AVF) creation process.

CKD is linked to immune activation associated with systemic low-grade inflammation and immunological insufficiency characterized by (i) monocytic expansion and decreased phagocytic activity associated with increased expression of TLRs and integrins and production of cytokines and reactive oxygen species (ROS); (ii) increased ROS production and apoptosis of circulating leukocytes; (iii) decreased CD4^+^/CD8^+^ T cell ratio and depletion of naïve and central memory T cells; (iv) deficiency of dendritic cells (DCs); (v) depletion of regulatory T cells (Tregs) and impaired immunoinhibitory activity; and (vi) depletion of B cells and impaired humoral immunity [16,17,18]. Therefore, inflammation contributes to the progression of CKD by enhancing the production and release of cytokines and adhesion molecules. Together, CKD-induced cytokines and adhesion molecules contribute to T cell adhesion and migration, subsequently attract profibrotic factors, and increase the development of vascular calcification and endothelial dysfunction [19].

Regardless of significant progress in understanding the role of vascular cells in immune responses modulating the pathogenesis of CKD-accelerated vascular inflammation and AVF maturation failure, whether CKD converts veins into immune endocrine organs remains unclear. Furthermore, low-throughput techniques used in current kidney and cardiovascular research laboratories limit our understanding of the role of veins in CKD. Therefore, high-throughput computational bioinformatics screens are often introduced to obtain an overview at the start of experimental projects. Initially, RNA-seq data can be profiled across various databases such as Gene Set Enrichment Analysis (GSEA, https://www.gsea-msigdb.org/gsea/index.jsp, accessed on 25 August 2022) and Ingenuity Pathway Analysis (IPA, https://www.nihlibrary.nih.gov/resources/tools/ingenuity-pathways-analysis-ipa, accessed on 15 August 2022). However, high-throughput differential gene expression analyses of veins from CKD patients and non-CKD controls are lacking, further contributing to our gaps in knowledge. Therefore, we hypothesized that CKD reprograms and transdifferentiates veins into a specialized immune endocrine organ, thereby accelerating vascular inflammation. To address this, we performed RNA-seq analysis of 48 upper-extremity veins used to create AVF and 20 non-CKD control veins and determined the gene expression changes of canonical and noncanonical secretome genes; innate immune response genes; plasma membrane proteins-encoding genes; metabolomic proteins-encoding genes; kinases; and transcription factors (TFs). Our finding show that CKD converts veins into an immune organ by upregulating 13 cytokines and chemokines genes, over 50 canonical and noncanonical secretomes, 12 innate immune response genes, and 18 cell membrane protein genes for increased intercellular communication. Furthermore, CKD changes the composition of immune cells in veins and upregulates CX3CR1 chemokine signaling. Additional changes in CKD veins include upregulation of genes involved in mitochondrial bioenergetics to induce immunometabolic reprogramming, over 20 fibroblast-enriched genes and fibrogenic factors, and numerous cell death/survival programs to shape the composition of immune cell types in veins, as well as reprogramming of protein kinase signal transduction pathways, including *SRPK3* and *CHKB*. The above changes may be related to upregulation of *MYCN*, *AP1*, and 11 other TFs known for their role in embryonic organ development, positive regulation of developmental growth, and muscle structure development. These results provide novel insights into the roles of veins as an immune endocrine organ and the effects of CKD in accelerating vascular inflammation and AVF maturation failure by upregulating secretomes and driving immune and vascular cell differentiation.

## 2. Materials and Methods

### 2.1. Study Subjects and Sample Collection

The CKD vein cohort consisted of 48 CKD patients who were undergoing surgery for AVF creation; 10 of them were reported here for the first time and 38 were described elsewhere [20,21]. Of these, 12 were classified as stage 5 CKD (CKD5; defined as an estimated glomerular filtration rate [eGFR] <15, not on dialysis), while 36 were hemodialysis dependent (ESRD). The patients provided written informed consent during their preoperative visit, under a protocol approved by the University of Miami Institutional Review Board and adherent to the Declaration of Helsinki (IRB protocol approval number 20110645). We obtained a 1–5 mm cross-section of the preaccess vein (45 basilic, 2 brachial, and 1 median-cubital) that would have been otherwise discarded after AVF creation. The non-CKD cohort included 20 organ donors whose tissues were donated for research purposes through a collaboration with the Life Alliance Recovery Agency. Cross-sectional samples of upper-extremity veins (18 basilic and 2 cephalic), approximately 2 cm in length, were obtained post-mortem following organ procurement procedures. All veins were collected in RNA*later* (QIAGEN, Germantown, MD, USA) and stored at –80 °C. Table 1 presents the demographic and clinical characteristics of both patient cohorts.

### 2.2. Total RNA Extraction and Sequencing

A 1 mm cross-section (~50–60 mg of tissue) was ground to a fine powder in a Spex/Mill 6770 cryogenic grinder (SPEX SamplePrep, Metuchen, NJ, USA). Total RNA was isolated with Trizol (Thermo Fisher Scientific, Waltham, MA, USA) and further purified using the E.Z.N.A. Total RNA Kit I (Omega Bio-tek, Norcross, GA, USA) as previously described [20,21]. Preparation and sequencing of RNA libraries were carried out in the John P. Hussman Institute for Human Genomics, Center for Genome Technology. Briefly, total RNA was quantified and qualified using the Agilent Bioanalyzer to have an RNA Integrity Score (RIN) > 5. A total of 500 ng of total RNA was used as input for the Illumina TruSeq Stranded Total RNA Library Prep Kit with Ribo-Zero (illumina, San Diego, CA, USA) to create ribosomal RNA-depleted sequencing libraries. Each sample had a unique barcode to allow for multiplexing and was sequenced to >40 million raw reads in a single-end 75-base-pair (bp) sequencing run on the Illumina NextSeq500. Raw sequence data were processed by the on-instrument real-time analysis software (v.2.4.11) to basecall files. These were converted to de-multiplexed FASTQ files with the Illumina-supplied scripts in the BCL2FASTQ software (v2.17). The quality of the reads was determined with the FASTQC software (http://www.bioinformatics.babraham.ac.uk/projects/fastqc, accessed on 1 July 2022) to evaluate per-base sequence quality, duplication rates, and overrepresented k-mers. Illumina adapters were trimmed from the ends of the reads using the Trim Galore! package (http://www.bioinformatics.babraham.ac.uk/projects/trim_galore, accessed on 1 July 2022). Reads were aligned to the human reference genome (hg38) with the STAR aligner (v2.5.2). Gene count quantification for total RNA was performed using the GeneCounts function within STAR against the GENCODE v25 human transcript.gtf file. The raw RNA-seq data are accessible in the NCBI Gene Expression Omnibus through the GEO accession numbers GSE119296, GSE220796, and GSE233264.

### 2.3. Bioinformatic Analyses and Annotation of Differentially Expressed Genes

Differentially expressed genes (DEGs) between CKD and non-CKD veins were identified in DESeq2 after adjusting for batch effects [22]. Transcripts with log2 (fold change) FC ≥ 1 or log2 FC ≤ −1 and false discovery rate (FDR)-adjusted *p* < 0.05 were considered significantly upregulated or significantly downregulated, respectively, in veins from CKD patients compared to non-CKD controls. Metascape (https://metascape.org/gp/index.html#/main/step1, accessed on 20 October 2022) was used for enrichment analysis. This website contains the core of most existing gene annotation portals. More details about Metascape can be found in the cited references [23]. As shown in Figure 1, functional annotation of genes was collected from six types of secretomic gene sets, including the Human Protein Atlas database-classified cytokines/cytokine receptors and chemokines (1249 genes) [24], the canonical secretome (2640 genes with signal peptide) [12], the caspase-1-dependent noncanonical secretome (964 genes), the caspase-4-dependent noncanonical secretome (1223 genes), and the exosome secretome (6560 genes) [25]. We also looked at the gene expression changes of 1367 innate immune response genes from the InnateDB database 2202 human plasma membrane protein genes (11% of all protein-coding human genes) from the HPA, immune cell markers and genes in 45 human immune cells from single-cell RNA sequencing of tissues from 12 deceased organ donors [26], HPA-classified endoplasmic reticulum (ER) protein encoding genes, 1136 human nuclear genome DNA-encoded mitochondrial genes, HPA-classified 81 glucose metabolism (glycolysis/gluconeogenesis) pathway genes, 53 tricarboxylic acid (TCA) cycle genes, 31 pentose phosphate pathways genes, 159 oxidative phosphorylation genes, and catabolic pathway genes, human metabolomic proteins from the human metabolome database, 959 fibrogenic genes in eight fibrotic diseases [27], four types of cell death, including apoptosis, necrosis, pyroptosis, cell death in response to oxidative stress, as well as efferocytosis, kinomes (a complete list of 621 kinases encoded in the human genome), and transcription factors (TFs), a complete list of 1496 human genome-encoded [22].

## 3. Results

### 3.1. Chronic Kidney Disease Changes the Transcriptome of Upper-Extremity Veins

Chronic kidney disease is characterized by a series of progressive vascular insults that ultimately results in various vascular pathologies and dysfunction. Transcriptional dysregulation of endothelial and mural cells in vessels is thought to underlie CKD-associated vasculopathy. However, there is little information about CKD-related changes in gene expression in human veins, despite their frequent use for vascular access creation. In this study, we compared the transcriptomic profiles of 48 veins from CKD/ESRD patients and 20 veins from non-CKD organ donors. The CKD group presented differences in baseline characteristics with respect to non-CKD donors that were inherent in this patient population. This included older age (56 ± 14 versus (vs.) 42 ± 15 years), a higher proportion of Black people/African Americans (52 vs. 10%), and a higher prevalence of hypertension (98 vs. 35%) and diabetes (54 vs. 15%) (Table 1).

Bioinformatic analyses uncovered a total of 1290 DEGs between groups (absolute log2 fold change > 1, FDR < 0.05), with 292 upregulated and 998 downregulated genes in CKD vs. non-CKD veins (Figure 2A). Differentially expressed genes were organized into three main clusters according to their normalized count distribution among individuals (Figure 2B). Clusters 1 (292 genes) and 3 (383 genes) best represented CKD-associated differences (Figure 2C).

### 3.2. Chronic Kidney Disease Transforms Veins into an Immune Organ by Upregulating 13 Cytokine and Chemokine Genes, 36 Canonical Secretomes, 2 Caspase-1- and 5 Caspase-4-Dependent Noncanonical Secretomes, and 28 Exosome Secretomes

Both canonical caspase-1 and noncanonical caspase-4/11 play a critical role in sensing pathogen-associated molecular patterns (PAMPs)/conditional damage-associated molecular patterns (DAMPs), mediating inflammatory cytokine secretion and cell death (pyroptosis), and accelerating CKD-promoted neointima hyperplasia [23,24,25,26,27,28] (Figure 3A). CKD-generated uremic toxins are conditional DAMPs or HAMPs, which may promote vascular smooth muscle phenotype switching [11,29]. We recently reported that the uremic toxin trimethylamine N-oxide (TMAO) produced by the gut microbiome synergizes with fungi and yeast-wall-derived β-glucan in promoting innate immune memory (also termed “trained immunity”) in aortic endothelial cells [14,30,31,32]. In addition, we previously reported that ESRD upregulates proinflammatory secretomes in peripheral blood mononuclear cells (PBMCs) and aortas [12,13]. Increased inflammation and cellular injury caused by canonical [26,27,28,33,34,35,36,37] and noncanonical pyroptosis [38] are strongly linked to the progression of diabetic kidney disease, aggravating renal fibrosis, glomerular sclerosis, and tubular injury [39]. However, it is unknown whether CKD modulates the transcriptomic changes in vein vascular cells and increases the secretion of proinflammatory cytokines and chemokines, the canonical secretomes (secretory proteins with signal peptides), the caspase-1-dependent noncanonical secretomes, the caspase-4-dependent noncanonical secretomes, and exosome secretomes [13]. We hypothesized that the expressions of cytokines and chemokines, as well as the canonical secretomes [12], and three types of noncanonical secretomes [40], including caspase-1-dependent, caspase-4-dependent, and exosome genes, are differentially modulated in vein vascular cells in CKD environment. To examine this hypothesis, we collected 1249 cytokines and their interactors (receptors) and chemokines genes from the HPA (https://www.proteinatlas.org/search/cytokine; https://www.proteinatlas.org/search/chemokine, accessed on 20 December 2022) as we have reported [12], and examined their expression levels in our RNA-seq data. As shown in Figure 3B and Appendix A, CKD significantly modulated the gene expression changes of 190 cytokines/cytokine receptors and chemokines. Among the 190 DEGs, CKD significantly upregulated 13 genes (6.84%) and downregulated 177 genes (93.15%). Metascape pathway analysis identified the top pathways of the 13 CKD-upregulated cytokines/cytokine receptor and chemokines, including regulation of cytokine production involved in immune response, NABA-matrisome-associated (genes encoding ECM-associated proteins including ECM-affiliated proteins, ECM regulators, and secreted factors https://www.gsea-msigdb.org/gsea/msigdb/geneset_page.jsp?geneSetName=NABA_MATRISOME_ASSOCIATED, accessed on 20 December 2022), cytokine–cytokine receptor interaction, innate immune response [41], and positive regulation of cytokine production (Figure 3C).

To gain a comprehensive understanding of whether CKD regulates the secretory functions of vein vascular cells, we collected 2640 canonical secretome genes from the HPA [12]; 964 caspase-1-dependent noncanonical secretome genes [42]; 1223 caspase-4-dependent noncanonical secretome genes [43]; and 6560 exosome secretome genes from a comprehensive exosome database (http://www.exocarta.org, accessed on 20 December 2022) [44], as we have reported [40]. As shown in Figure 3D, CKD significantly modulated the gene expression changes of 226 canonical secretomes. Among the 226 differentially modulated canonical secretome genes, 36 genes (15.9%) were significantly upregulated, and 190 genes (84.1%) were downregulated in the veins of CKD patients (Appendix A). In addition, CKD modulated the gene expression changes of 18 caspase-1-dependent noncanonical secretomes with two upregulated genes and 16 downregulated genes (Appendix A); and CKD differentially modulated the gene expression changes of 56 caspase-4-dependent noncanonical secretomes with 5 (8.92%) upregulated genes and 51 (91.07%) downregulated genes (Appendix A). Furthermore, 144 genes of exosome secretomes were differentially modulated by CKD, with 28 (19.44%) significantly upregulated genes and 116 (80.55%) downregulated genes (Appendix A). The top pathways of the 36 CKD-upregulated canonical secretome genes (Figure 3E) and the 28 CKD-upregulated exosome secretome genes (Figure 3F) were revealed by Metascape pathway analysis. These data indicate that CKD transforms veins into a specialized immune–endocrine organ by upregulating the expression of 13 cytokines/cytokine receptors and chemokines genes, and over 50 secretome genes including 36 canonical secretome genes, two caspase-1-dependent noncanonical secretome genes, 5 caspase-4-dependent noncanonical secretome genes, and 28 exosome secretome genes (Figure 3G), which are secretory immune effectors and further affect the pathophysiological functions and inflammatory and immune responses of the kidney and other organs and tissues (Figure 3H) via autocrine, paracrine, and endocrine manners [13].

### 3.3. Chronic Kidney Disease Increases the Innate Immune Response by Upregulating 12 Innate Immune Response Genes and 18 Plasma Membrane Protein Genes in the Veins for Increased Intercellular Communication

The secreted cytokines and chemokines, and canonical and noncanonical secretomes, play significant roles in promoting the innate immune response in vascular immune cells [22,40,45,46]. As we previously reported and reviewed, after a brief exposure to endogenous or exogenous insults, innate immune cells can develop an exaggerated immunological response and a long-term inflammatory phenotype. This results in an altered response to a second challenge after the return to a nonactivated state. This is referred to as “trained immunity” (also termed innate immune memory) [32,36,47,48,49,50], and the cytokine TNF-α serves as a readout [14]. We hypothesized that one of the mechanisms underlying the gene upregulation of cytokines and secretomes is the enhanced innate immune response. To test this hypothesis, we examined the expression changes of a comprehensive list of 1615 innate immune response genes (innatome) [41,49] in our RNA-seq of CKD veins compared to non-CKD control veins. Our data analysis showed that CKD significantly modulates the expression of 119 innate immune response genes. Among the 119 differentially modulated genes, 12 genes (10.08%) were significantly upregulated, and 107 genes (89.9%) were downregulated in the veins of CKD patients (Figure 4A and Appendix A). Metascape pathway analysis of the 12 CKD-upregulated innate immune response genes showed the top pathways, including immune response-activating cell surface receptor signaling pathway, regulation of cytokine production involved in immune response, and positive regulation of cytokine production (Figure 4B).

The SET domain containing lysine methyltransferase 7 (SETD7, also called SET7/9) was the first lysine methyltransferase identified to specifically monomethylate lysine-4 histone 3 (H3K4me1), a marker for transcriptional activation [51,52]. SET7 is a key regulator of trained immunity [13,53]. We hypothesized that the expression of CKD-upregulated genes will be inhibited when SET7 is downregulated. To test our hypothesis, we examined the overlapped genes between CKD-upregulated genes and SET7 depletion-significantly downregulated genes collected from the NCBI GEO database (https://www.ncbi.nlm.nih.gov/geo/query/acc.cgi?acc=GSE53038, accessed on 15 January 2023) [54]. We found that 15 CKD-upregulated genes were significantly downregulated by SET7 deficiency (Figure 4C), and the top pathways involved are NABA secreted factors (genes encoding secreted soluble factors, https://www.gsea-msigdb.org/gsea/msigdb/geneset_page.jsp?geneSetName=NABA_SECRETED_FACTORS, accessed on 15 January 2023), growth, embryonic organ development, hemostasis, and regulation of growth (Figure 4D).

The detection of PAMPs by a variety of host pattern recognition receptors (PRRs) leads to the release of proinflammatory cytokines and chemokines, which are essential for an effective innate immune response [55,56,57]. The infected or dying cells have the ability to transmit PAMPs and host PRR signaling proteins to uninfected bystander cells by a variety of plasma membrane proteins to bypass pathogen evasion strategies and potentiate innate immune forward, reverse, or bidirectional signaling [49,58]. This bystander activation of innate immunity represents an alternative strategy for the host to control infections through cell-to-cell (intercellular) communication [58,59,60] (Figure 5A). We hypothesized that plasma membrane proteins are upregulated in CKD to potentiate the innate immune response. To test this hypothesis, we collected 2202 human plasma membrane proteins coding genes from the HPA (https://www.proteinatlas.org/search/subcell_location%3APlasma+membrane%2CCell+Junctions, accessed on 15 December 2022), which contains 11% of all protein-coding human genes. Our data analysis showed that CKD significantly modulates the gene expression changes of 73 human plasma membrane proteins. Among 73 differentially modulated cell/plasma membrane protein coding genes, 18 genes (24.66%) were significantly upregulated (Figure 5B), and 55 genes (75.34%) were downregulated in CKD veins (Appendix A). The top pathways of the 18 CKD-upregulated plasma membrane protein coding genes were revealed by Metascape pathway analysis (Figure 5C). This analysis indicates that CKD increases intra- and extra-vein intercellular communication to potentiate immune responses (Figure 5D).

### 3.4. Chronic Kidney Disease Reshapes the Composition of the Immune System in the Veins and Upregulates CX3CR1 Chemokine Signaling

The immune system is traditionally categorized into innate and adaptive compartments, and multiple interactions between these compartments exist. A direct and nonspecific response to infection and tissue damage is provided by the innate immune system. The key cellular elements of the innate immune system are monocytes, macrophages, dendritic cells (DCs), natural killer (NK) cells, endothelial cells [47,49,61], and vascular smooth muscle cells [28,29] as we proposed. Innate immune cells express various molecular PRRs, such as TLRs, which enable the cells to respond to bacterial and viral proteins and fragments of damaged cells [62]. On the other hand, the adaptive immune system has evolved to respond to threats to the host in a highly specific way [63]. The adaptive immune cells include B and T lymphocytes, which respond to specific antigens expressed by pathogens [22,64]. Both innate and adaptive immune cells have a memory function [58,65] that facilitates a faster and stronger immune response when pathogens are re-encountered [14,32,50]. CKD adversely affects the innate and adaptive cellular immune systems [66,67]. The number and function of each cell type are impacted to varying degrees. However, the effect of CKD on the major innate and adaptive immune cell populations in the veins has not been examined. We hypothesized that CKD modulates innate and adaptive immune cell compositions in the veins. Therefore, we examined the markers and top-upregulated genes in 45 human immune cell subsets from single-cell RNA-seq data [68] deposited in the ArrayExpress database at EMBL-EBI (www.ebi.ac.uk/arrayexpress, accessed on 15 December 2022) under accession number E-MTAB-11536. As shown in Figure 6A, CKD significantly modulates the expression of 4 out of 10 markers and top upregulated genes in two monocyte subsets and upregulated CX3CR1 and downregulated three genes; CKD differentially modulates the expression of 11 out of 27 markers and top upregulated genes in six macrophage subsets and upregulated CX3CR1 and downregulated 10 genes; CKD differentially modulates the expression of 3 out of 21 markers and top upregulated genes in three DCs subsets and upregulated CX3CR1 and downregulated 2 genes; CKD differentially modulates the expression of 5 out of 20 markers and top upregulated genes in three NK cells subsets and upregulated CX3CR1 and downregulated 4 genes; CKD differentially modulates the expression of 5 out of 39 markers and top upregulated genes in 18 T cell subsets and upregulated CX3CR1 and downregulated 4 genes; CKD differentially modulates the expression of 5 out of 27 markers and top upregulated genes in 11 B cell subsets and only downregulated 5 genes.

The chemokine receptor C-X3-C motif receptor 1 (CX3CR1) is highly expressed in various immune cells, including monocytes, macrophages, DCs, T cells, and NK cells [69,70]. As shown in Figure 6B, immune cells expressing CX3CR1 play critical roles in the pathology of certain human tissues. CX3CR1 expression in peripheral blood mononuclear cells increases leukocyte migration to inflammation sites [70]; however, in monocytes, macrophages, DCs, and T cells, it increases cytokine secretion, ROS production [71], extracellular matrix formation, foam cell formation, vascular remodeling, oxidative stress, and inflammation [69]. Furthermore, CKD is associated with the depletion of immune cells, including T cells, B cells, and DCs, leading to impaired humoral and cellular immunity and impaired antigen presentation [17]. Collectively, these results demonstrate that CKD increases the infiltration of inflammatory cell types but decreases the number of adaptive immune cell types in veins.

### 3.5. Chronic Kidney Disease Upregulates Five Endoplasmic Reticulum Protein-Coding Genes and Three Mitochondrial Genes, Impairs Mitochondrial Bioenergetics, and Induces Immunometabolic Reprogramming in Veins

Endoplasmic reticulum (ER) stress is involved in a variety of pathogenic conditions, which include the development of CKD. We recently reported that uremic toxins and CKD induce ER and mitochondrial stress in endothelial cells, PBMCs, and kidney tissues [14]. Therefore, we hypothesized that CKD affects the expression of ER genes in veins. To examine this hypothesis, we collected 539 ER protein-encoding genes, which include 3% of all protein-coding human genes that encode proteins that localize to the ER from the HPA (https://www.proteinatlas.org/search/subcell_location%3AEndoplasmic+reticulum, accessed on 5 February 2023). Our analysis showed that CKD upregulates 5 ER protein-encoding genes in veins (Figure 7A) and downregulates 24 ER protein-encoding genes (Appendix A).

Mitochondria are considered sources and targets of uremic toxins [72]. Experimental evidence suggests that CKD is associated with impaired mitochondrial metabolism. Animal studies demonstrate that CKD promotes deficits in mitochondrial electron transport and pyruvate dehydrogenase activity and abnormalities in specific enzymes and metabolic pathways related to mitochondrial energy generation [73,74]. We hypothesized that CKD modulates the transcription of genomic (nuclear) DNA-encoded mitochondrial genes (mitocarta genes) and induces impaired mitochondrial function and bioenergetics in human veins. To test this hypothesis, we collected 1136 human mitochondrial genes (mitocarta) from the Broad Institute (https://www.broadinstitute.org/mitocarta/mitocarta30-inventory-mammalian-mitochondrial-proteins-and-pathways, accessed on 20 December 2022) [46,75]. We found that CKD significantly modulated the expression of 19 human mitochondrial genes, among which 3 mitochondrial genes were significantly upregulated (Figure 7A) and 16 mitochondrial genes were downregulated in human veins (Appendix A).

The main function of mitochondria is oxidative phosphorylation for the generation of cellular ATP, but they also play an important role in multiple metabolic pathways. In addition to their well-known catabolic role, they have a critical anabolic role in providing the carbon skeletons for the biosynthesis of glucose, fatty acids, and amino acids [76]. Therefore, we examined the expression changes of oxidative phosphorylation pathway genes, the TCA cycle genes, the pentose phosphate pathway genes, and glucose-metabolism-related genes, as well as catabolic genes downloaded from the HPA. Our data analysis showed that CKD significantly upregulated one TCA cycle gene and two glucose-metabolism-related genes. However, CKD significantly downregulated 9 oxidative phosphorylation genes, 2 pentose phosphate pathway genes, and 15 glucose-metabolism-related genes (Figure 7B). In addition, CKD significantly upregulated 6 and downregulated 110 catabolic genes (Figure 7C and Appendix A).

Furthermore, we examined the expression changes of human small molecule metabolite-related enzymes, transporters, and other molecule genes downloaded from the Human Metabolome Database (HMDB) (https://hmdb.ca, accessed on 20 December 2022) [77] and found that CKD significantly modulates the expression of 355 human small molecule metabolite-related enzymes, transporters, and other metabolism genes, among which 39 genes were significantly upregulated (Figure 7D) and 316 genes were downregulated (Appendix A) in the vein of CKD patients. In addition, Metascape pathway analysis showed the top pathways of the 39 CKD-upregulated metabolomic genes (Figure 7E). Taken together, the results demonstrated that CKD promoted ER stress, mitochondrial dysfunction, impaired mitochondrial bioenergetics, and increased immunometabolic reprogramming in veins (Figure 7F).

### 3.6. Chronic Kidney Disease Transdifferentiates Vein Fibroblasts by Upregulating 20 Fibroblast Genes and Reprograms the Fibrogenic Process in Veins by Upregulating 6 Fibrogenic Genes, Priming the CKD Vein for AVF Failure and Multiple Organ Failure

Fibroblasts are thought to be the major matrix-producing cells of the kidney and are therefore clinically relevant as key mediators of renal fibrosis associated with progressive renal failure. In addition to their structural function in extracellular matrix synthesis, they also play a critical role in the response to a tissue injury, such as the immune response and wound healing [78]. We hypothesized that CKD modifies fibroblast gene expression and reprograms the fibrogenic process in veins. To test this hypothesis, we collected 401 fibroblast-related genes from the HPA (https://www.proteinatlas.org/search/cell_type_category_rna%3AFibroblasts%3BCell+type+enriched%2CGroup+enriched%2CCell+type+enhanced+AND+show_columns%3Atissuespecificity+AND+sort_by%3Atissue+specific+score, accessed on 15 January 2023) [79]. As shown in Figure 8A, CKD significantly modulates the expression of 52 fibroblast genes, with 18 (34.62%) upregulated and 34 (65.38%) downregulated genes in the veins. In addition, Metascape pathway analysis showed that the 18 CKD-upregulated fibroblast genes have five functions, including NABA ECM glycoproteins matrisome (genes encoding structural ECM glycoproteins, https://www.gsea-msigdb.org/gsea/msigdb/geneset_page.jsp?geneSetName=NABA_ECM_GLYCOPROTEINS, accessed on 20 December 2022), ossification, NABA ECM affiliated (genes encoding proteins affiliated structurally or functionally to extracellular matrix proteins, https://www.gsea-msigdb.org/gsea/msigdb/geneset_page.jsp?geneSetName=NABA_ECM_AFFILIATED, accessed on 20 December 2022), cellular response to transforming growth factor beta stimulus, and regulation of hormone levels (Figure 8B). Furthermore, we examined the expression changes of 959 common fibrosis genes identified in eight different fibrotic diseases, including renal fibrosis, hepatic fibrosis, lung fibrosis, heart fibrosis, intestinal fibrosis, pancreatic fibrosis, eye fibrosis, and skin fibrosis [80]. Figure 8C and Appendix A show that CKD significantly upregulated 6 and downregulated 95 fibrogenic genes out of 101 CKD-differentially modulated fibrogenic genes. Metascape pathway analysis showed that six CKD-upregulated fibrogenic genes have different signaling pathways, including response to wounding, cellular response to transforming growth factor beta stimulus, and cell activation (Figure 8D). Collectively, these results demonstrate that CKD modulates the expression of fibroblast genes and upregulates 18 genes; it also reprograms the fibrogenic process in veins by upregulating six fibrosis genes to prime CKD-veins for AVF failure [81,82] and multiple organ failure.

### 3.7. Chronic Kidney Disease Reprograms Several Cell Death and Survival Programs and Promotes Programmed Cell Death (Apoptosis) More Than Other Types of Cell Death

As many as 18 cell death types have been characterized, including intrinsic apoptosis, extrinsic apoptosis, mitochondrial permeability transition (MPT)-driven necrosis, necroptosis, ferroptosis, pyroptosis, parthanatos, entotic cell death, NETotic cell death, lysosome-dependent cell death, autophagy-dependent cell death, immunogenic cell death, cellular senescence, mitotic catastrophe, panoptosis, autosis, oxeiptosis, and alkaliptosis [33,83,84,85,86,87]. We hypothesized that CKD reprograms cell death in veins. To test this hypothesis, we collected 1992 programmed cell death (apoptosis)-related genes (http://www.informatics.jax.org/go/term/GO:0012501, accessed on 15 January 2022), 68 necrotic cell-death-related genes (http://www.informatics.jax.org/go/term/GO:0070265, accessed on 15 January 2022), 34 pyroptosis-related genes (http://www.informatics.jax.org/go/term/GO:0070269, accessed on 15 January 2022), and 102 oxidative-stress-induced cell-death-related genes (http://www.informatics.jax.org/go/term/GO:0070269, accessed on 15 January 2022), as well as 47 apoptotic cell clearance (efferocytosis)-related genes (http://www.informatics.jax.org/vocab/gene_ontology/GO:0043277, accessed on 15 January 2022). As shown in Figure 9A, CKD significantly upregulated 6 and downregulated 145 programmed cell death (apoptosis)-related genes (Appendix A). Notably, five genes out of six (83.33%) CKD-upregulated programmed cell death (apoptosis)-related genes, including cysteine-rich protein 1 (*CRIP1*), pleiotrophin (*PTN*), glutamate Ionotropic receptor NMDA type subunit 2A (*GRIN2A*), transforming growth factor beta 2 (*TGFB2*), and frizzled related protein (*FRZB*), play a role in the positive regulation of apoptotic process and function as proapoptotic genes. However, only one gene (*CX3CR1)* plays a role in the negative regulation of the apoptotic process and functions as an antiapoptotic gene. Interestingly, among the 145 CKD-downregulated apoptosis genes, 69 genes were negative regulators of the apoptotic process (antiapoptosis), 16 genes were both negative and positive regulators of the apoptosis process (pro- and antiapoptosis), and 35 genes were positive regulators of the apoptosis process (proapoptosis) (Appendix A). Furthermore, we examined the expression changes of the necrotic cell-death-related genes and found that CKD did not upregulate but did significantly downregulate three necrotic cell-death-related genes (Figure 9B). In addition, we examined the expression changes of the pyroptosis-related genes and found that CKD did not upregulate pyroptosis-related genes but did significantly downregulate five pyroptosis-related genes. It is worth noting that all of the pyroptosis genes that were downregulated by CKD were promoters of pyroptosis (Figure 9C).

CKD also induces changes in the expression of oxidative-stress-induced cell death genes (Figure 9D). There were only 15 downregulated genes in the veins of CKD patients. Among those 15 downregulated genes, 8 genes were identified as positive regulators of oxidative-stress-induced cell death, and 6 genes were identified as negative regulators of oxidative-stress-induced cell death. Finally, we examined the gene expression changes of apoptotic cell clearance (efferocytosis) genes in the veins of CKD patients. Our data showed that CKD significantly downregulated nine efferocytosis genes, and all these genes function as positive regulators of apoptotic cell clearance (Figure 9E). Collectively, our analysis revealed that (i) CKD upregulated more proapoptotic genes and downregulated more antiapoptotic genes in veins, suggesting that CKD promotes apoptotic cell death in veins; (ii) CKD only downregulated necrotic cell death genes in veins; (iii) CKD only downregulated pyroptosis genes in veins; (iv) CKD downregulated eight positive regulators and six negative regulators of oxidative-stress-induced cell death genes, suggesting that CKD regulates oxidative-stress-induced cell death; (v) CKD induced more apoptosis and oxidative-stress-induced cell death other than necrosis and pyroptosis; and (vi) CKD downregulated efferocytosis genes, suggesting that CKD impaired efferocytosis.

### 3.8. Chronic Kidney Disease Reprograms Protein Kinase Signal Transduction Pathways and Upregulates SRPK3 and CHKB; CKD Reprograms Vein Transcriptomes and Upregulates MYCN, AP1, and 11 Other Transcription Factors in Veins

Protein kinases and phosphatases are enzymes that catalyze the transfer of a phosphate group to a protein. Phosphorylation and dephosphorylation are important posttranslational modifications of native proteins (Figure 10A). These biological processes play important roles in intracellular signal transduction cascades and switching enzymatic activities [14,88,89]. In CKD, protein kinases such as serine/threonine mammalian target of rapamycin (mTOR) and the AMP-activated protein kinase (AMPK) pathway play a unique role in signal transduction pathways, energy metabolism, inflammation, stress, and cell death in the kidneys [90,91,92]. However, an important question remained about whether CKD affects kinase pathways in the veins. We hypothesized that CKD modulates the expression of kinases in veins. To test this hypothesis, we examined the expression changes of the total kinome (a complete list of 661 kinases encoded in the human genome) [93]. As shown in Figure 10B, CKD modulated the gene expression changes of 47 kinases, with only two significantly upregulated kinase-encoding genes, including Serine/Arginine-Rich Protein-Specific Kinase 3 (*SRPK3*) and Choline Kinase Beta (*CHKB*). In addition, 45 kinase-encoding genes were significantly downregulated in CKD veins. Of note, *SRPK3* is involved in muscle development, NADPH oxidase 2 (NOX2)-regulated ischemia-reperfusion-induced myocardial injury [94], and the immune response, and is alternatively spliced among muscle types, indicating muscle-specific regulation [82,95]. *CHKB* encodes the choline kinase beta enzyme. This enzyme plays a key role in phospholipid biosynthesis and catalyzes the first step in phosphatidylcholine and phosphatidylethanolamine synthesis, two crucial lipids for cellular membranes, including those in mitochondria [96]. *CHKB* mutations have been linked to mitochondrial deficiencies and other disorders. *CHKB* knockdown reduced phosphatidylcholine and choline kinase activity. The impairment of choline/phosphatidylcholine kinase activity leads to changes in the composition of phospholipids in the mitochondrial membrane, resulting in a disorder involving the structure and function of mitochondria [97,98]. These results demonstrated that CKD reprograms kinomes and upregulates *SRPK3* and *CHKB* pathways in veins.

Transcription factors (TFs) are master regulators of fundamental biological processes due to their ability to regulate the expression of multiple gene targets. Their expression and functions can be regulated by transcriptional and posttranslational mechanisms in response to endogenous or exogenous changes in the environment [99]. TFs can drive cell differentiation [100] as well as de- and transdifferentiation [101]. Many human diseases are caused by mutations in TFs and TF-binding sites. Previous studies have shown that TF deregulation is associated with the development or progression of kidney disease [102]. We hypothesized that CKD differentially modulates the gene expression of a set of specific TFs. To test this hypothesis, we collected 1496 TFs from the HPA (https://v20.proteinatlas.org/search/protein_class:transcription+factors accessed on 16 May 2023), as we previously reported [12]. As shown in Figure 11A, CKD modulates the gene expression of 50 TFs, among which 13 TFs (26%) were significantly upregulated, including the FosB proto-oncogene, AP-1 transcription factor subunit (*FOSB*), D-box Binding PAR BZIP transcription factor (*DBP*), MYCN proto-oncogene, BHLH transcription factor (*MYCN*), zinc finger protein 837 (*ZNF837*), Retinoid X receptor gamma (*RXRG*), HLF transcription factor, PAR BZIP family member (*HLF*), Distal-less homeo box 5 (*DLX5*), Zinc finger protein 835 (*ZNF835*), SRY-box transcription factor 15 (*SOX15*), REST corepressor 2 (*RCOR2*), Forkhead box S1 (*FOXS1*), Zinc finger protein 285 (*ZNF285*), and GLI family zinc finger 1 (*GLI1*). In contrast, CKD significantly downregulated 37 TFs genes (74%) in veins (Appendix A). Metascape pathway analysis in Figure 11B shows that the 13 CKD-upregulated TFs had three significant pathways, including embryonic organ development, positive regulation of developmental growth, and muscle structure development. The *FOSB* is an AP-1 transcription factor, which contributes to the regulation of as many as 1577 target genes (containing AP-1 binding sites in the promoter of genes; https://maayanlab.cloud/Harmonizome/gene_set/AP-1/MotifMap+Predicted+Transcription+Factor+Targets, accessed on 17 February 2023), including cytokines, chemokines, and secretome genes. To examine whether the AP-1 target genes are upregulated in CKD veins, we used the Venn diagram and found that CKD significantly upregulated 10 AP-1 target genes in veins (Figure 11C). Metascape pathway analysis showed the three top pathways of the CKD-upregulated AP-1 target genes, including negative regulation of monoatomic ion transport, muscle contraction, and regulation of system process (Figure 11D). Furthermore, the MYCN transcription factor has 8482 target genes (https://maayanlab.cloud/Harmonizome/gene_set/MYC/CHEA+Transcription+Factor+Targets, accessed on 17 February 2023), which contribute significantly to the expression of cytokines, chemokines, immune genes, and vascular reprogramming genes [103]. Therefore, we used the Venn diagram to examine whether the MYC target genes are upregulated in CKD veins. Our data showed that CKD significantly upregulated 42 MYC target genes in veins (Figure 11E), which include the TF FOSB. Metascape pathway analysis showed the top pathways of the CKD-upregulated MYC target genes (Figure 11F). Since FOSB is a target for the MYC, this indicates that MYC is upstream of FOSB. These data indicate that uremic toxins and CKD stimulate vein cells, reprogram vein transcriptomes, and upregulate 13 transcription factors for inflammation [104], which modulate prograde and retrograde signaling between the nucleus and other cytoplasmic organelles such as mitochondria in response to uremic toxins/CKD stimulation (Figure 11G), as we reported [105].

## 4. Discussion

Chronic kidney disease has steadily increased over the past decades as a result of an aging population and metabolic disorders [106]. Patients with CKD have increased cardiovascular risk and a decreased estimated glomerular filtration rate (eGFR). The adverse cardiovascular effects of CKD are also emerging rapidly [107]. Major vascular events further increase healthcare costs in patients with varying degrees of CKD severity [108]. In addition, loss of renal function is often irreversible and leads to ESRD. Despite advances in CKD care, slowing renal progression remains a challenge, and a new focus is urgently needed.

Arteriovenous fistula (AVF) is the first choice and the preferred approach for creating hemodialysis vascular access in ESRD, as its use has been associated with better survival than other means of access [109]. Despite significant improvements in preoperative patient assessment and surgical planning, vascular access failure remains a common and unpredictable complication in hemodialysis patients. AVF maturation failure due to vein damage is the leading cause of morbidity and mortality in those patients. Preexisting vascular pathologies associated with CKD, including neointima hyperplasia (NIH) resulting from smooth muscle cells (SMCs) combined with matrix deposition, vascular calcification, and fibrosis, may result in stenosis and, ultimately, occlusion [110]. Uremic toxin accumulation, chronic inflammation, and oxidative stress have been identified as CKD-specific alterations that increase cardiovascular risk. The association between CKD and cardiovascular mortality is clearly influenced by vascular damage, particularly atherosclerosis and vascular calcification (VC), and uremia can also impair the patency of the AVF [111,112,113].

Upper-extremity veins are important in ESRD because they may be used to create vascular accesses [114]. However, unlike the arteries, the contribution of CKD-related factors to chronic venous remodeling and transcriptomic changes in CKD veins has been poorly studied. To our knowledge, this is the first study to combine whole-transcriptomic RNA sequencing with new knowledge-based bioinformatics analysis [46,115] of vein samples from CKD patients compared to non-CKD controls. Our first major finding is that CKD turns veins into an immune–endocrine organ by upregulating cytokines, chemokines, and secretome genes in veins. The second major finding is that CKD promotes specific innate immune responses by upregulating innate immune (innatome) genes, modulating plasma membrane protein-mediated intercellular communication, reshaping the composition of immune cells, and upregulating *CX3CR1* chemokine signaling in veins. The third major finding is that CKD upregulates ER genes and mitochondrial genes, induces mitochondrial dysfunction, and increases bioenergetic and immunometabolic reprogramming. The fourth major finding is that CKD reprograms the fibrogenic process in veins by transdifferentiating vein fibroblasts and upregulating fibroblast genes, as well as reprograms the fibrogenic process in veins by upregulating fibrotic genes, and thus primes the CKD vein for AVF failure and multiple organ failure. The fifth major finding is that CKD reprograms cell death and survival programs and induces more apoptosis than other types of cell death. Finally, CKD reprograms kinomes and upregulates serine/arginine-rich protein-specific kinase 3 (*SRPK3*) and choline kinase beta (*CHKB*) pathways in veins, and CKD reprograms vein transcriptomes and upregulates *MYCN*, *AP1*, and 11 other transcription factors. Of note, a lot of gene categories including cytokines and chemokines (via their receptors), plasma membrane proteins, which initiate plasma membrane signaling, TFs, and cell death genes in this manuscript, are checkpoints that warrant additional studies in the future. To validate the gene expression changes in CKD veins, we crossed our independent RNA-seq dataset of HAECs treated with uremic toxin TMAO [14] with our vein RNA-seq dataset and found that 19 genes were shared between the 1290 CKD differentially modulated genes in veins and the 369 TMAO differentially modulated genes.

To summarize our findings here, we propose a new working model (Figure 12). In traditional lymphoid organs, when exposed to antigens, naïve lymphocytes and other immune cells undergo development, maturation, activation, and differentiation/polarization. During an infection, antigen-presenting cells including dendritic cells and macrophages ingest pathogens (antigens) and present antigen epitopes to T cells [116,117,118] via membrane–protein interaction and intercellular communication. In addition to presenting antigens, they provide costimulatory signals [58] and secretome cytokine signaling (autocrine and paracrine functions), resulting in increased cytokine, chemokine, and secretome secretion and increased inflammatory immune response. In contrast, in CKD, uremic toxins including cytokines and other secretomes activate immune cells in the wall of veins. Once these cells become activated, they communicate through membrane–protein interactions to transfer signals between each other and amplify inflammatory signaling through the activation of the MYCN/AP-1 signaling pathway. In addition, these activated immune cells enter the blood circulation and move to other organs (long-distance endocrine) [119], leading to increased cytokine, chemokine, and secretome secretion, increasing inflammatory immune response, and potentially inducing cell proliferation. 

One limitation of this study is that it only relies on the RNA-seq and, similar to all the RNA-seq data analyses that due to the low-throughput nature of current verification techniques in every laboratory, including ours, we could not verify the result we found with the analyses of high-throughput data, which are similar to all the studies with RNA-seq, single-cell RNA-seq, metabolomics, chromatin immunoprecipitation (CHIP)-seq, and other-omics data. We acknowledge that carefully designed in vitro and in vivo experimental models will be needed in the future to verify the gene expression changes we report here. Another limitation is the huge difference in age between CKD patients and non-CKD controls (56 vs. 42). Unfortunately, these are inherent differences between CKD patients and organ donor populations. Since the biological age is not the same as chronological age, it is not possible to control for this potential confounding effect. In the future, when the age-matched veins from two populations are available, we will verify the findings reported in this manuscript. Of note, we confirmed that 270/292 genes upregulated in CKD with respect to non-CKD veins remained significant (log2fold change > 1, FDR < 0.05) after adjusting for age in DESeq2 analyses, as well as 916/998 downregulated genes. These results indicate that global differences between the CKD vein transcriptomes and non-CKD transcriptomes are not only age-related.

## Figures and Tables

**Figure 1 cells-12-01482-f001:**
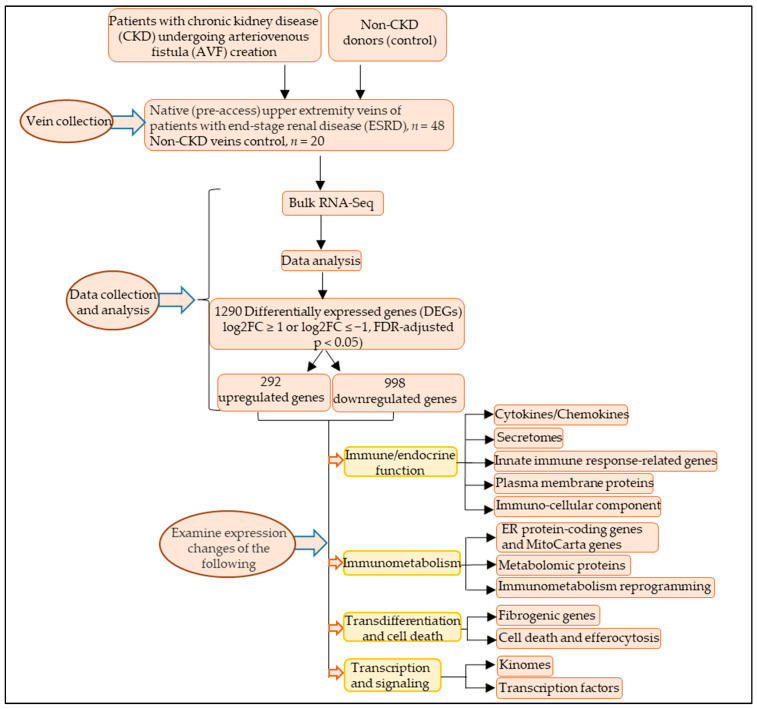
Differentially expressed 12 groups of trained immunity and immunometabolic reprogramming genes were examined. Flow chart of the study. Native (preaccess) upper-extremity veins were collected from patients with chronic kidney disease (CKD) undergoing 2-stage arteriovenous fistula (AVF) surgery (*n* = 48) and veins from non−CKD donors (*n* = 20) were used as a control. Vein samples were analyzed using RNA sequencing (RNA-seq). Comparing the veins of CKD patients to those of controls, 292 genes were significantly upregulated, and 998 genes were significantly downregulated (log2FC ≥ 1 or log2FC ≤ −1, FDR-adjusted *p* < 0.05). The gene expression changes of 12 groups of genes, including cytokine/cytokine receptors and chemokines, canonical and noncanonical secretomes, innate immune response-related genes, human plasma membrane proteins, immunocellular components, ER protein-coding genes and mitocarta genes, human metabolomic proteins, immunometabolic reprogramming genes, fibroblast and fibrogenic genes, cell death and efferocytosis-related genes, kinomes, and transcription factors were examined.

**Figure 2 cells-12-01482-f002:**
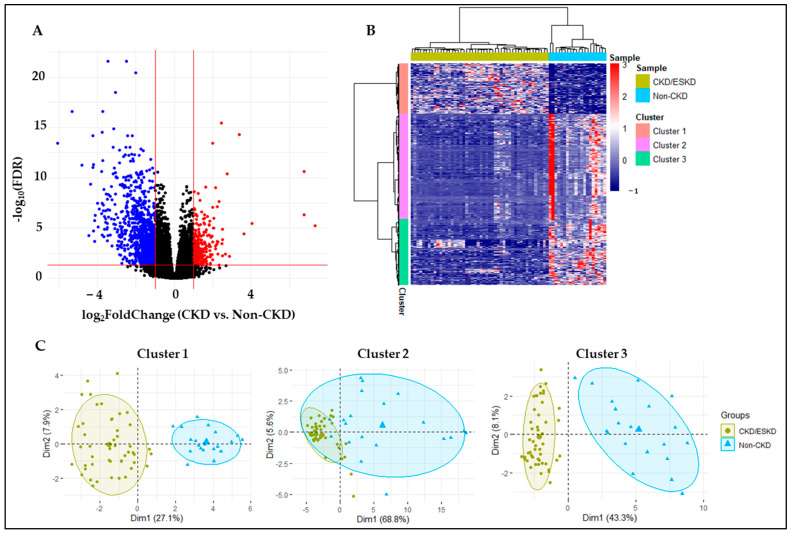
The transcriptomic profiles of 48 veins from chronic kidney disease (CKD)/end-stage renal disease (ESRD) patients and 20 veins from non-CKD organ donors. (**A**) A volcano plot analysis showing the DEGs (red dots representing the upregulated genes and blue dots representing the downregulated genes) in the veins of CKD patients and non-CKD controls. (**B**) Differentially expressed genes are organized into three main clusters according to their normalized count distribution among individuals. (**C**) Clusters 1 (292 genes) and 3 (383 genes) best represent CKD-associated differences.

**Figure 3 cells-12-01482-f003:**
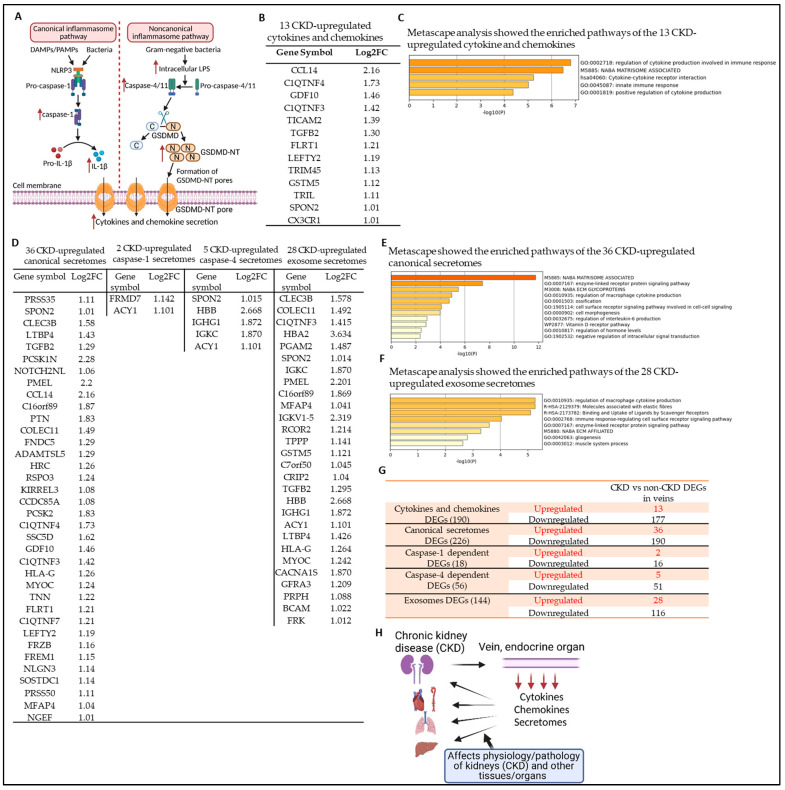
Chronic kidney disease (CKD) transforms veins into an immune organ by upregulating 13 cytokine and chemokine genes, 36 canonical secretome genes, 2 caspase-1-dependent and 5 caspase-4-dependent noncanonical secretome genes, and 28 exosome secretome genes. (**A**) A schematic diagram shows the canonical and noncanonical inflammatory caspase pathways and cytokine and chemokine secretion. (**B**) The gene expression changes of cytokines/cytokine receptors and chemokines in the veins of CKD patients compared to those of non-CKD controls. CKD significantly modulates the expression of 190 cytokine/cytokine receptors and chemokine genes in veins. Among 190 CKD-modulated genes, 13 genes were significantly upregulated, and 177 genes were downregulated in the veins of CKD patients. The genes listed in Appendix A were downregulated by CKD. (**C**) Metascape pathway analysis showed the top pathways of the 13 CKD-upregulated cytokine/cytokine receptors and chemokine genes. The 1249 cytokines/cytokine receptors and chemokines gene list was obtained from the HPA (https://www.proteinatlas.org/search/cytokine; https://www.proteinatlas.org/search/chemokine, accessed on 20 December 2022). (**D**) The gene expression changes of 2640 canonical secretomes (signal-peptide-mediated exocytic secretory pathway), 964 caspase-1-dependent noncanonical secretomes (nonsignal-peptide-mediated), 1223 caspase-4-dependent noncanonical secretomes, and 6560 exosome secretomes in the veins of CKD patients compared to those of non-CKD controls. CKD significantly modulates the expression of 226 canonical secretome genes with 36 upregulated genes and 190 downregulated genes shown in Appendix A; 18 caspase-1-dependent noncanonical secretomes with two upregulated genes and 16 downregulated genes shown in Appendix A; 56 caspase-4-dependent noncanonical secretomes with 5 upregulated genes and 51 downregulated genes shown in Appendix A, and 144 exosome secretomes with 28 upregulated genes and 116 downregulated genes shown in Appendix A. The 2640 canonical secretome gene list was downloaded from the HPA (https://www.proteinatlas.org, accessed on 20 December 2022), 964 noncanonical caspase-1-dependent noncanonical secretome gene list was generated from PMID: 18329368, 1223 noncanonical caspase-4-dependent noncanonical secretome gene list was extracted from PMID: 28196878, and 6560 exosome secretome gene list was downloaded from a comprehensive exosome database (http://www.exocarta.org, accessed on 20 December 2022). (**E**) Metascape pathway analysis showed the top pathways of the 36 CKD-upregulated canonical secretome genes. (**F**) Metascape pathway analysis showed the top pathways of the 28 CKD-upregulated exosome secretome genes. (**G**) A summary of CKD differentially expressed canonical, caspase-1-dependent, caspase-4-dependent, and exosome secretome genes. (**H**) A schematic diagram shows that in a CKD environment, the vein has an endocrine function to secrete cytokines/chemokines and secretomes, which have effects on the physiological and pathological functions of the kidney (CKD) and other tissues and organs.

**Figure 4 cells-12-01482-f004:**
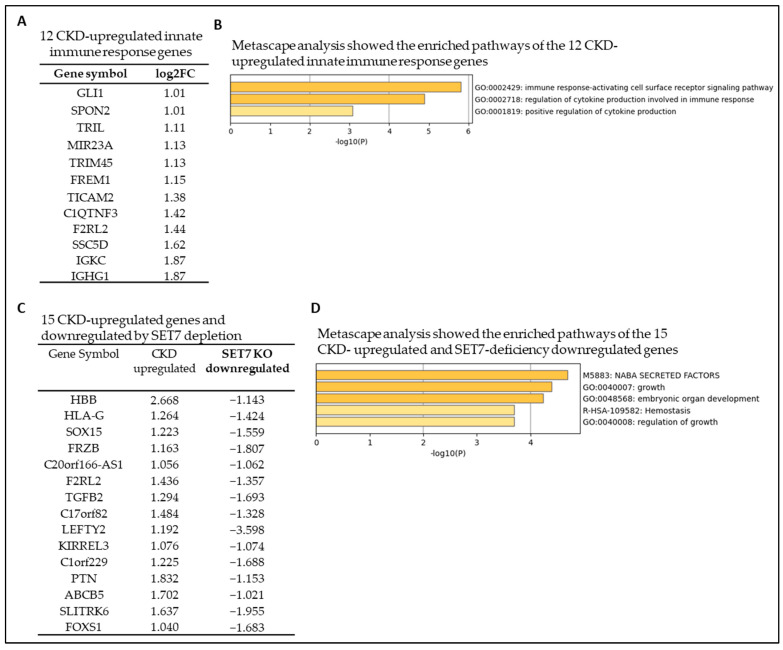
Chronic kidney disease (CKD) promotes specific innate immune responses and upregulates 12 innate immune (innatome) genes. The gene expression changes of the innate immune response genes in the veins of CKD patients are compared to those of non-CKD controls. (**A**) CKD significantly modulates the expression of 119 innate immune response genes with 12 upregulated genes in the veins of CKD patients. (**B**) Metascape pathway analysis showed top pathways of 12 CKD-upregulated immune response genes. (**C**) The trained immunity regulator SET7 contributes to CKD. Microarray expression data from SET7-depleted H9 human embryonic stem cells (GSE53038, PMID: 25479749) were collected (Log2FC > 1, *p* < 0.05). The genes upregulated by CKD were crossed with the genes downregulated by SET7 depletion. Fifteen genes upregulated in CKD were significantly downregulated in SET7 depletion. (**D**) Metascape analysis showed top pathways of 15 CKD-upregulated and SET7 deficiency-downregulated genes. The 1367 innate immune response gene list was collected from the InnateDB database (https://www.innatedb.com, accessed on 20 December 2022).

**Figure 5 cells-12-01482-f005:**
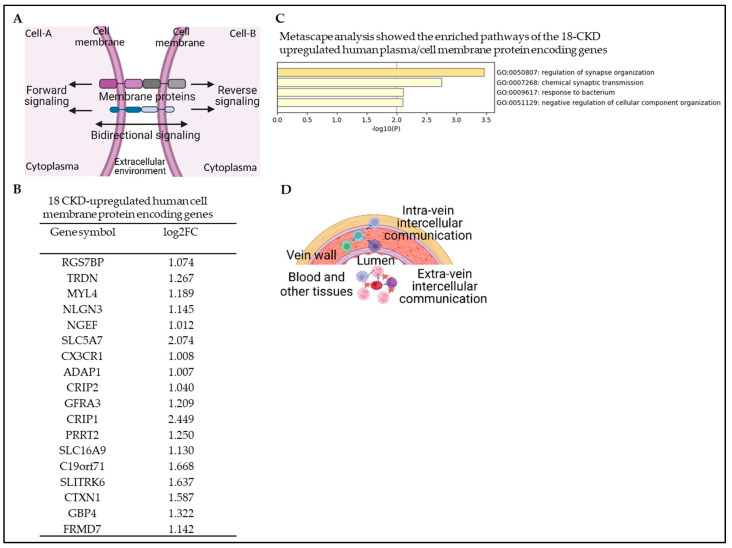
Chronic kidney disease (CKD) increases immune responses by modulating plasma membrane protein-mediated intercellular communication. The gene expression changes of 2202 human plasma membrane protein genes (11% of all protein-coding human genes) from the HPA (https://www.proteinatlas.org/humanproteome/subcellular/plasma+membrane, accessed on 15 December 2022) in the veins of CKD patients were compared to those of non-CKD controls. (**A**) A schematic diagram shows the intercellular communication through plasma membrane proteins and the forward, reverse, and bidirectional signaling. (**B**) CKD modulates the gene expression of 73 human plasma membrane proteins with 18 upregulated genes and 55 downregulated genes in veins. (**C**) Metascape pathway analysis showed the top pathways of the 18 CKD-upregulated plasma membrane proteins. (**D**) A schematic diagram depicts the intercellular communication between intravenous and extravenous (blood and other tissues). CKD downregulated genes are listed in Appendix A.

**Figure 6 cells-12-01482-f006:**
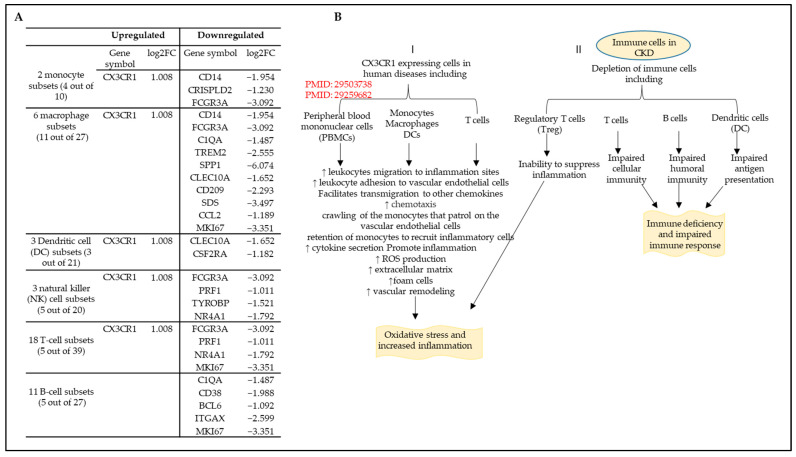
Chronic kidney disease (CKD) reshapes the composition of immune cells and upregulates CX3CR1 chemokine signaling. The cellular composition and human immune cell subsets change in the veins of CKD patients. A total of 45 immune cell groups, including 2 monocyte subsets, 6 macrophage subsets, 3 DCs subsets, 3 NK cell subsets, 18 T cell subsets, and 11 B cell subsets, were collected from the single−cell RNA sequencing of tissues from 12 deceased organ donors (PMID: 35549406). (**A**) CKD upregulated CX3CR1 and downregulated 3 monocyte genes, 10 macrophage genes, 2 DC genes, 4 NK cell genes, 4 T cell genes, and 5 B cell genes. (**B**) A summary of CKD’s effect on immune cells (innate and adaptive) and the resulting effect on inflammation and the immune response.

**Figure 7 cells-12-01482-f007:**
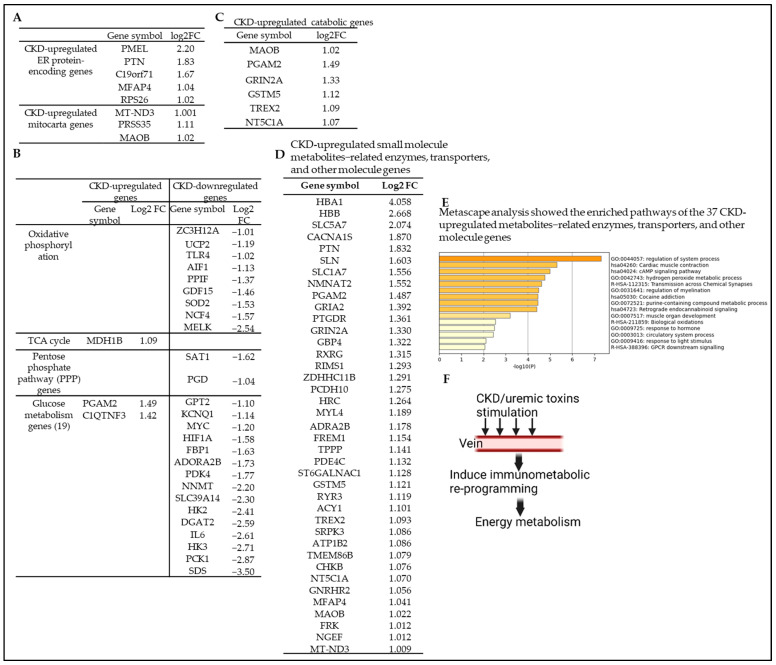
Chronic kidney disease (CKD) upregulates 5 endoplasmic reticulum (ER) genes and 3 mitochondrial genes, induces mitochondrial dysfunction, and increases bioenergetic and immunometabolic reprogramming. The gene expression changes of ER protein-encoding genes and human nuclear genome DNA-encoded mitochondrion genes (mitocarta) in the veins of CKD patients were compared to those of non-CKD controls. (**A**) CKD significantly upregulated 5 ER protein-encoding genes and 3 mitocarta genes and downregulated 24 ER protein-encoding genes and 16 mitocarta genes in the veins of CKD patients. The ER protein-encoding genes were downloaded from the HPA (https://www.proteinatlas.org/humanproteome/subcellular/endoplasmic+reticulum, accessed on 5 February 2023) and mitocarta genes from the Broad Institute (https://www.broadinstitute.org/mitocarta/mitocarta30-inventory-mammalian-mitochondrial-proteins-and-pathways, accessed on 20 December 2022). (**B**) The gene expression changes of energy metabolism pathways in the veins of CKD patients were compared to those of non-CKD controls. CKD significantly downregulated 9 oxidative phosphorylation genes, 2 pentose phosphate pathway genes, and 15 glucose metabolism genes and upregulated one TCA cycle gene and two glucose metabolism genes. The glucose metabolism (glycolysis/gluconeogenesis) pathway genes, TCA cycle genes, pentose phosphate pathway genes, oxidative phosphorylation genes, and catabolic pathway genes were downloaded from the HPA. (**C**) CKD significantly upregulated 6 and downregulated 110 catabolic genes. The CKD-downregulated catabolic genes are listed in Appendix A. (**D**) The gene expression changes of human small molecule metabolites-related enzymes, transporters, and other molecule genes in the veins of CKD patients were compared to those of non-CKD controls. CKD significantly modulates the expression of 355 genes; among them, 39 genes were significantly upregulated, and 316 genes were downregulated in the veins of CKD patients. The 316 significantly downregulated genes were listed in Appendix A. (**E**) Metascape pathway analysis showed the top pathways of the 39 CKD-upregulated metabolomic proteins. The human small molecule metabolites-related enzymes, transporters, and other molecule genes were downloaded from the human metabolome database (HMDB) (https://hmdb.ca, accessed on 20 December 2022). (**F**) A schematic diagram shows that CKD/uremic effects on the vein affect the cellular immunometabolic reprogramming required for energy metabolism in the cells.

**Figure 8 cells-12-01482-f008:**
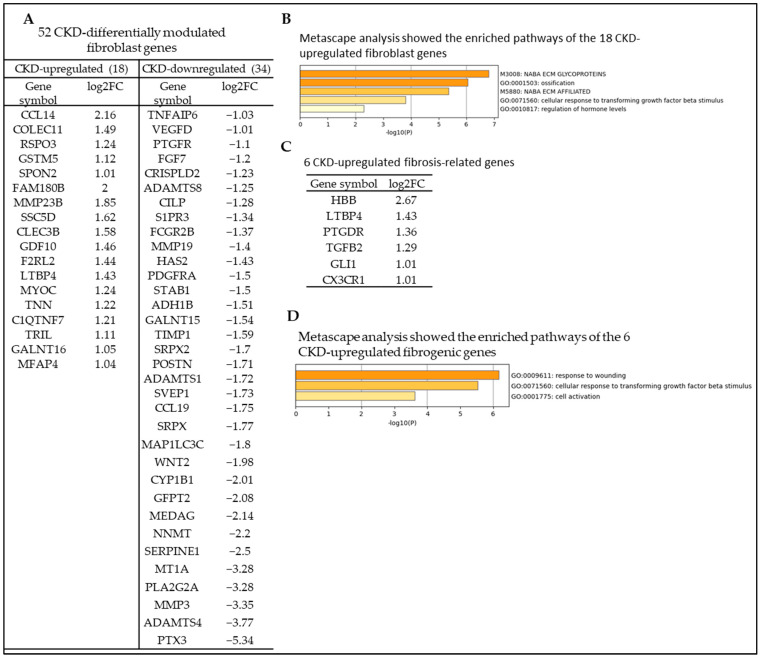
Chronic kidney disease (CKD) reprograms the fibrogenic process in veins by transdifferentiating vein fibroblasts and upregulating 18 fibroblast genes and reprograms the fibrogenic process in veins by upregulating 6 fibrotic genes and primes CKD vein for AVF failure and multiple organ failure. (**A**) The gene expression changes of the fibroblast genes in the veins of CKD patients compared to non-CKD controls. CKD upregulated 18 and downregulated 34 out of 52 differentially modulated fibroblast genes in veins. The 461 fibroblast genes were collected from the HPA. (**B**) Metascape pathway analysis showed the top pathways of the 18 CKD-upregulated fibroblast genes. (**C**) The gene expression changes of the fibrogenic genes in the veins of CKD patients compared to non-CKD controls. CKD upregulated 6 and downregulated 95 genes out of 101 differentially modulated fibrosis-related genes in veins. The 959 fibrogenic genes in eight fibrotic diseases were collected from previous publication (PMID: 33519923). (**D**) Metascape analysis showed the top pathways of the 6 CKD upregulated fibrogenic genes.

**Figure 9 cells-12-01482-f009:**
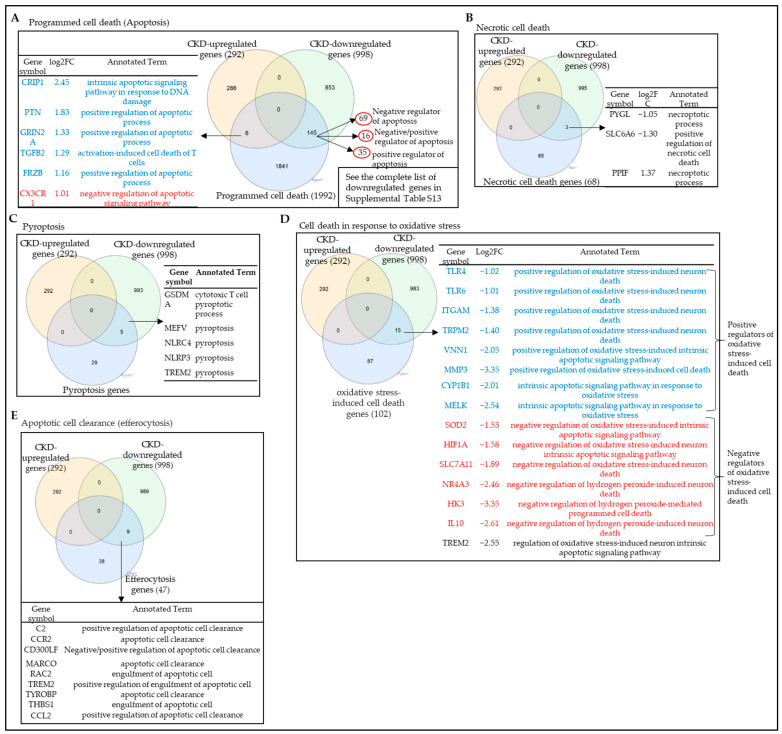
Chronic kidney diseases (CKD) reprogram several cell death and survival programs and promote programmed cell death (apoptosis) more than other types of cell death. The gene expression changes of 1992 programmed cell death, 68 necrotic cell death, 34 pyroptosis, 102 oxidative-stress-induced cell death, and 47 efferocytosis in the veins of CKD patients were compared to those of non-CKD controls. (**A**) A Venn diagram reveals that CKD significantly upregulated the expression of 6 and downregulated the expression of 145 programmed cell death (apoptosis)-related genes in the veins. The programmed cell death (apoptosis) gene list was downloaded from the Gene Ontology http://www.informatics.jax.org/go/term/GO:0012501/15/1/2023 (accessed on 16 May 2023). (**B**) A Venn diagram reveals that CKD significantly downregulated the expression of 3 necrotic cell death-related genes in the veins. The necrotic cell death gene list was downloaded from the Gene Ontology http://www.informatics.jax.org/go/term/GO:0070265/15/01/2023 (accessed on 16 May 2023). (**C**) A Venn diagram reveals that CKD significantly downregulated the expression of 5 pyroptosis genes in the veins. The pyroptosis gene list was downloaded from the Gene Ontology http://www.informatics.jax.org/go/term/GO:0070269/15/01/2023 (accessed on 16 May 2023). (**D**) A Venn diagram reveals that CKD significantly downregulated the expression of 15 oxidative-stress-induced cell death-related genes in the veins. The oxidative-stress-induced cell death-related gene list was downloaded from the Gene Ontology http://www.informatics.jax.org/go/term/GO:0036473/15/01/2023 (accessed on 16 May 2023). (**E**) A Venn diagram reveals that CKD significantly downregulated the expression of 9 efferocytosis-related genes in the veins. The efferocytosis-related gene list was downloaded from the Gene Ontology http://www.informatics.jax.org/vocab/gene_ontology/GO:0008219/15/01/2023 (accessed on 16 May 2023).

**Figure 10 cells-12-01482-f010:**
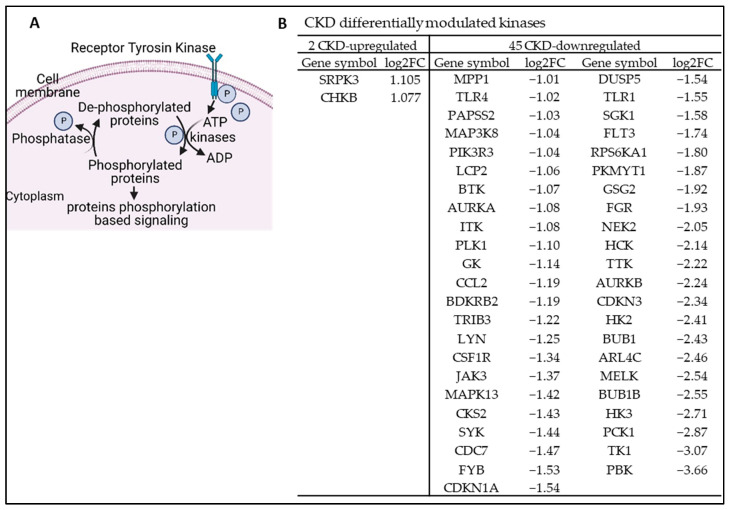
Chronic kidney disease (CKD) reprograms protein kinase signal transduction pathways and upregulates Serine/Arginine-Rich Protein-Specific Kinase 3 (*SRPK3*) and Choline Kinase Beta (*CHKB*) pathways in veins. The gene expression changes of the kinome (a complete list of 661 kinases encoded in the human genome) in the veins of CKD patients were compared to those of non-CKD controls. (**A**) A schematic diagram showed protein kinases and phosphatases. Protein kinases and phosphatases regulate biological signals by acting as enzymes that catalyze phosphorylation and dephosphorylation in biological organisms. Phosphorylation catalyzed by protein kinases retransmits incoming signals by activating the protein kinases’ substrates. The signal transmission will be terminated by the dephosphorylation (inactivation) of their substrates, a reaction catalyzed by protein phosphatases. (**B**) CKD modulates the gene expression changes of 47 kinomes with 2 upregulated genes, including *SRPK3* and *CHKB*. The 661 kinases or kinase-related genes were extracted from PMID: 31289205.

**Figure 11 cells-12-01482-f011:**
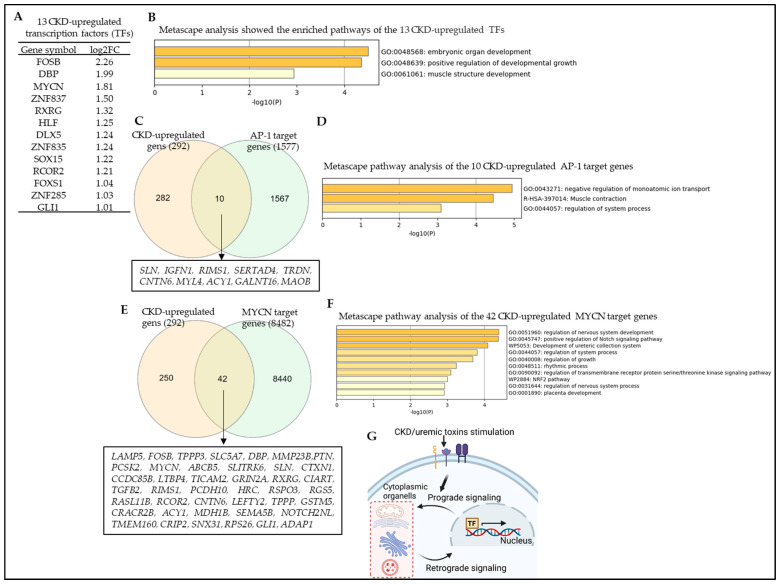
Chronic kidney disease (CKD) reprograms vein transcriptomes and upregulates MYCN, AP1, and 11 other transcription factors. The gene expression changes of transcription factors (TFs) in the veins of CKD patients were compared to those of non-CKD controls. The 1496 TFs were collected from the HPA (https://www.proteinatlas.org/search/protein_class:Transcription+factors, accessed on 17 February 2023). (**A**) CKD modulates the gene expression changes of 50 TFs with 13 upregulated genes and 37 downregulated genes in the vein of CKD patients (see Appendix A). (**B**) Metascape pathway analysis showed the top pathways of the 13 CKD-upregulated TFs. (**C**) CKD upregulated 10 AP-1 target genes, which were collected from the MotifMap dataset: https://maayanlab.cloud/Harmonizome/gene_set/AP-1/MotifMap+Predicted+Transcription+Factor+Targets, accessed on 17 February 2023. (**D**) Metascape pathway analysis revealed the top pathways of the 10 CKD-upregulated AP-1 target genes. (**E**) CKD upregulated 42 MYC target genes, which were collected from the MotifMap dataset https://maayanlab.cloud/Harmonizome/gene_set/MYC/CHEA+Transcription+Factor+Targets, accessed on 17 February 2023. (**F**) Metascape pathway analysis revealed the top pathways of the 42 CKD-upregulated MYC target genes. (**G**) A schematic diagram shows the prograde and retrograde signaling between the nucleus and cytoplasmic organelles.

**Figure 12 cells-12-01482-f012:**
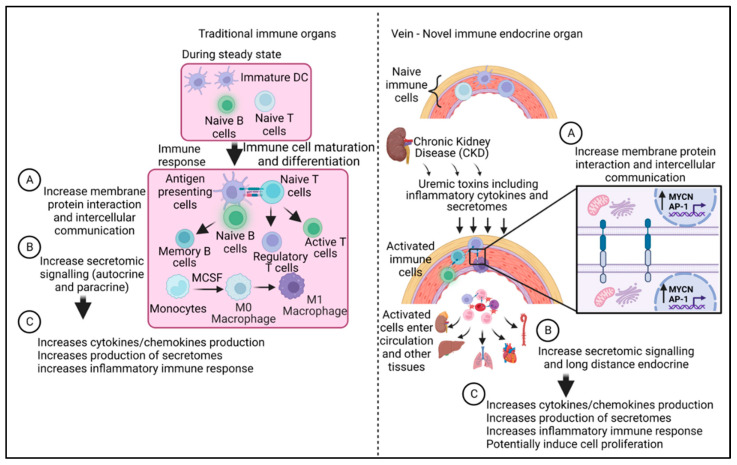
Our working model. In traditional lymphoid organs, when exposed to antigens, naïve lymphocytes and other immune cells undergo development, maturation, activation, and differentiation/polarization. During an infection, antigen-presenting cells, including dendritic cells and macrophages, ingest pathogens (antigens) and present them to T cells via membrane–protein interaction and intercellular communication. In addition to presenting antigens, they provide costimulatory signals and secretome cytokine signaling (autocrine and paracrine functions), resulting in increased cytokine/chemokine and secretome secretion and an increased inflammatory immune response. In contrast, in CKD, uremic toxins, including cytokines and other secretomes, activate immune cells in the wall of veins. Once these cells become activated, they communicate through membrane–protein interactions to transfer signals between each other and amplify inflammatory signaling through the activation of the MYCN/AP-1 signaling pathway. In addition, these activated immune cells enter the blood circulation and move to other organs (long-distance endocrine) leading to increased cytokine/chemokine and secretome secretion, increasing inflammatory immune response, and potentially inducing cell proliferation.

**Table 1 cells-12-01482-t001:** Baseline characteristics of the patient cohorts.

	CKD/ESRD	Non-CKD
No. of Donors	48	20
Age (mean ± SD)	56 ± 14	42 ± 15
Females *n* (%)	24 (50)	10 (50)
Race/Ethnicity (%)		
Black *n* (%)	25 (52)	2 (10)
Hispanic *n* (%)	17 (35)	7 (35)
White *n* (%)	6 (13)	11 (55)
Comorbidities (%)		
Hypertension *n* (%)	47 (98)	7 (35)
Diabetes *n* (%)	26 (54)	3 (15)
CAD *n* (%)	10 (21)	3 (15)
Stage 5 CKD n (%)	12 (25)	-
ESRD *n* (%)	36 (75)	-
Vein Type		
Basilic vein *n* (%)	45 (94)	18 (90)
Brachial vein *n* (%)	2 (4)	-
Cephalic vein *n* (%)	-	2 (10)
Median Cubital vein *n* (%)	1 (2)	-

Abbreviations: CKD: chronic kidney disease, ESRD: end-stage renal disease, CAD: coronary artery disease.

## Data Availability

The RNA sequencing data presented in this study can be found in the NCBI Gene Expression Omnibus through the GEO accession numbers GSE119296, GSE220796, and GSE233264.

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
