# Peer review of "Chronic Kidney Disease Transdifferentiates Veins into a Specialized Immune–Endocrine Organ with Increased MYCN-AP1 Signaling"

_cells, 2023, doi:10.3390/cells12111482_

Round 1
Reviewer 1 Report
Authors performed an extensive whole-transcriptomic RNA sequencing and bioinformatics analysis to show the differentiation of vein in CKD. The study was well performed and has provided plentiful novel gene alterations in CKD veins, compared with non-CKD ones. Comments are as the followings.
1. Authors may consider making some parts of this manuscript (Introduction, and Results) more concise. Readers may understand better with ease, knowing that the whole story is complex.
2. On the contrary, authors can address some more in the discussion. For example,
Among these many altered gene expression patterns, are there any “check-point” candidates that warrant extensive studies?
The present method grounded the whole vein. Therefore, it will be not possible to know the cell types (endothelial、muscle、adventitia?) that contribute to the gene alterations.
Authors mentioned that uremic toxins might activate the immunological responses in the vascular cells and the activated immune cells and secreted cytokines might exert a long-distance endocrine effect. However, the vice versa phenomenon might also exist as well. Readers understand that it’s a complex interplay, but not a one-way effect.
3. Can authors describe and suggest some future prospects, based on your findings?
Author Response
Reviewer 1
Comments and Suggestions for Authors
Authors performed an extensive whole-transcriptomic RNA sequencing and bioinformatics analysis to show the differentiation of vein in CKD. The study was well performed and has provided plentiful novel gene alterations in CKD veins, compared with non-CKD ones.
We greatly appreciate the reviewer for these supportive comments.
Comments are as the followings.
- Authors may consider making some parts of this manuscript (Introduction, and Results) more concise. Readers may understand better with ease, knowing that the whole story is complex.
We greatly appreciate the reviewer for the suggestion. We have made our efforts to make it more concise and understandable.
- On the contrary, authors can address some more in the discussion. For example,
Among these many altered gene expression patterns, are there any “check-point” candidates that warrant extensive studies?
Thanks a lot for your comments. We have added the following paragraph to the discussion section “A lot of gene categories in this manuscript are check-point, including plasma membrane proteins, which initiate plasma membrane signaling, cytokines and chemokines (via their receptors), transcription factors (master regulators, TFs), and cell death genes that warrant additional studies in the future”.
The present method grounded the whole vein. Therefore, it will be not possible to know the cell types (endothelial、muscle、adventitia?) that contribute to the gene alterations.
Thanks a lot for raising this point. Currently, we are preparing a few manuscripts about single cell RNA-sequencing. Therefore, in the near future we will report gene expression analysis on the single-cell RNA sequencing by cell clusters in the veins of CKD patients versus non-CKD controls.
Authors mentioned that uremic toxins might activate the immunological responses in the vascular cells and the activated immune cells and secreted cytokines might exert a long-distance endocrine effect. However, the vice versa phenomenon might also exist as well. Readers understand that it’s a complex interplay, but not a one-way effect.
Thank you for the insightful comment. We fully agree that uremic toxins might activate immunological responses and then the cell type actually have a two-way street. In 2018, we reported that expression of uremic toxin generation enzymes can be modulated in metabolic diseases (PMID: 28930551). In 2023, we showed that gut microbiota-generated uremic toxin trimethylamine N-oxide (TMAO) generating enzyme flavin monooxygenase (FMO3) can be expressed in aorta cells (PMID: 36394956). Once the vascular immune cells are activated, they secrete cytokines and chemokines into the blood circulation and exert long-distance endocrine effect on other immune and non-immune cells.
- Can authors describe and suggest some future prospects, based on your findings?
Thanks for pointing this out. Based on our findings, in the future, we will use the single cell RNA sequencing to analyze the gene expression changes in the veins by cell clusters (we have added this to the discussion section).
Reviewer 2 Report
Saaoud et al present an original research paper entitled "Chronic Kidney Disease Transdifferentiates Veins into a Specialized Immune-Endocrine Organ with Increased MYCN-AP1 Signaling ". In this paper, they rely on RNA-seq data form a cohort of CKD and non CKD patients. They conclude that in CKD, uremic toxins including cytokines and other secretomes activate immune cells in the wall of veins. Once these cells become activated, they communicate through membrane protein interactions to transfer signals between each other and amplify inflammatory signaling through the activation of the MYCN/AP-1 signaling pathwayThe authors claim that the patients provided written informed consent during their preoperative visit, under a protocol approved by the University of Miami Institutional Review Board. Can they please give a specifc number?
the authors used sections of 45 basilic, 2 brachial, and 1 median-cubital. Can they justify the use of the 3 laters?
the age difference between the 2 groups is huge (56 vs 42) ; I think it is a major confusing factor. Can the authors comment on it? CKD is an accelerated ageing process. So here they compare younger people without CKD to older people with CKD.
The major drawback of this study is that it only relies on the RNA-Seq and has no independant validation. I would suggest to treat HUVEC cells with the TMAo uremic toxin and see if the same set of genes are altered.
Author Response
Reviewer 2
Comments and Suggestions for Authors
Saaoud et al present an original research paper entitled "Chronic Kidney Disease Transdifferentiates Veins into a Specialized Immune-Endocrine Organ with Increased MYCN-AP1 Signaling ". In this paper, they rely on RNA-seq data form a cohort of CKD and non-CKD patients. They conclude that in CKD, uremic toxins including cytokines and other secretomes activate immune cells in the wall of veins. Once these cells become activated, they communicate through membrane protein interactions to transfer signals between each other and amplify inflammatory signaling through the activation of the MYCN/AP-1 signaling pathway.
The authors claim that the patients provided written informed consent during their preoperative visit, under a protocol approved by the University of Miami Institutional Review Board. Can they please give a specifc number?
Thank you. The University of Miami IRB protocol approval number is 20110645. We have included this information under Study Subjects and Sample Collection in the manuscript.
the authors used sections of 45 basilic, 2 brachial, and 1 median-cubital. Can they justify the use of the 3 laters?
Thanks a lot. We used those upper extremity veins because those veins are commonly used for vascular access creation for dialysis in CKD patients and to understand the impact of CKD on the veins that may be used later on for improvement of either graft or arteriovenous fistula (AVF) creation.
the age difference between the 2 groups is huge (56 vs 42) ; I think it is a major confusing factor. Can the authors comment on it? CKD is an accelerated ageing process. So here they compare younger people without CKD to older people with CKD.
We agree with the reviewer on this insightful comment. Unfortunately, these are inherent differences between CKD patients and organ donor populations. It is also true that biological age is not the same as chronological age, so it is not possible to control for this potential confounding effect. In the future when the age-matched veins from two populations are available, we will verify the findings reported in this manuscript (this is one of the limitations of this study and we have added this to the Discussion section). Of note, we confirmed that 270/292 genes upregulated in CKD with respect to non-CKD veins remained significant (log2fold change>1, FDR<0.05) after adjusting for age in DESeq2 analyses, as well as 916/998 downregulated genes. These results indicate that global differences between the CKD vein transcriptomes and non-CKD transcriptomes are not only age-related. We have made these clarifications in the Results.
Reviewer 3 Report
This manuscript is written well, so I recommend to publish.
However this will need to revise just a little bit.
1) What is this research's hypothesis? Could you show this exactlly?
2) There are many figures in this manuscript, but this is not easy to see and understand, so could you revise these again?
3) Are there any other study limitations here?
Author Response
Reviewer 3
Comments and Suggestions for Authors
This manuscript is written well, so I recommend to publish.
However this will need to revise just a little bit.
- What is this research's hypothesis? Could you show this exactlly?
Thank you. The key finding listed in the title of our manuscript is a version of our hypothesis, so we write the title in a question sentence. We have included our hypothesis in the last paragraph of the introduction “
- There are many figures in this manuscript, but this is not easy to see and understand, so could you revise these again?
We have carefully revised the manuscript and make the figures clearer.
3) Are there any other study limitations here?
We greatly appreciate the reviewer for this suggestion. We have added the following paragraph to the discussion section “ One limitation of the current study similar to all the RNA-seq data analyses is that due to the low-throughput nature of current verification techniques in every laboratory, including ours, we could not verify the results we found with the analyses of high-throughput data, which are similar to all the -Omics studies with RNA-seq, single-cell RNA-seq, metabolomics, chromatin immunoprecipitation (CHIP)-seq, and other-Omics data. We acknowledge that carefully designed in vitro and in vivo experimental models will be needed in the future to verify the gene expression changes we reported here. Another limitation is the huge difference in age between CKD patients and non-CKD controls (56 vs. 42). Unfortunately, these are inherent differences between CKD patients and organ donor populations. Since the biological age is not the same as chronological age, so it is not possible to control for this potential confounding effect. In the future when the age-matched veins from two populations are available, we will verify the findings reported in this manuscript. Of note, we confirmed that 270/292 genes upregulated in CKD with respect to non-CKD veins remained significant (log2fold change>1, FDR<0.05) after adjusting for age in DESeq2 analyses, as well as 916/998 downregulated genes. These results indicate that global differences between the CKD vein transcriptomes and non-CKD transcriptomes are not only age-related”.
Round 2
Reviewer 2 Report
Changes are ok